# Phytochemicals and Monensin in Dairy Cows: Impact on Productive Performance and Ruminal Fermentation Profile

**DOI:** 10.3390/ani15152172

**Published:** 2025-07-23

**Authors:** Lucas Gonzalez-Chappe, Maria A. Bruni, Aline C. Dall-Orsoletta, Pablo Chilibroste, Ana Meikle, Maria L. Adrien, Alberto Casal, Juan P. Damián, Hugo Naya, Marisela Arturo-Schaan, Diego A. Mattiauda

**Affiliations:** 1Facultad de Agronomía, Universidad de la República, Av. Garzón 780, Montevideo 12900, Uruguay; 2CCPA France, CCPA Group, ZA du Bois de Teillay, 35150 Janzé, France; 3Facultad de Veterinaria, Universidad de la República, Ruta 8 km 18, Montevideo 12400, Uruguay

**Keywords:** transition period, ruminal fermentation, milk yield, protozoa

## Abstract

The use of antibiotics in livestock farming has raised significant public concern and is increasingly being questioned, particularly in the case of monensin, one of the most widely used additives. As a result, there is growing interest in finding alternatives to improve animal health and production efficiency. Phytochemicals are promising natural compounds with greater public acceptance; however, limited information is available regarding their application in dairy grazing systems during the transition period. This study demonstrated that the use of a phytochemical mixture enhanced the ruminal fermentation profile over time, which was associated with changes in the microbial community. In primiparous cows, both phytochemicals and monensin led to increased milk yield, indicating that the phytochemical mixture can achieve comparable performance to monensin.

## 1. Introduction

Dairy farms aim to enhance milk yield (MY) and quality while prioritizing animal welfare and sustainability. One approach is the strategic use of different dietary additives to manipulate ruminal fermentation to improve performance and efficiency while reducing environmental impact [1,2]. Monensin, a widely used additive in dairy cattle diets, enhances production and feed efficiency by increasing MY and reducing dry matter intake (DMI) [2,3,4]. Despite its benefits, the use of monensin as a growth promoter has been discontinued in the European Union [5], primarily due to regulatory changes aimed at phasing out antibiotic use in livestock production. While monensin is not used in human medicine, its classification as an ionophore and public concerns about the use of antimicrobials in food production have contributed to its restricted use. Consequently, the industry has sought alternative strategies to preserve feed efficiency and animal health in dairy systems, with natural products emerging as a more consumer-acceptable option [6].

Research and commercialization of plant-derived bioactive compounds in animal production have gained significant interest due to their positive effects on the health and productivity of ruminants [7,8]. These compounds exhibit antimicrobial properties similar to those of monensin, as well as antioxidant properties, and are promising feed additives capable of enhancing rumen fermentation, improving feed efficiency, and promoting animal health [7]. Phytochemical supplementation has shown the ability to alter ruminal fermentation by modulating the ruminal microbiota in both in vitro and in vivo studies [1,9]. However, the impact of phytochemicals on volatile fatty acid (VFA) production has been inconsistent, with reports ranging from negligible effects to both positive and negative outcomes [1,10]. Notably, phytochemicals have demonstrated a capacity to reduce ruminal ammonia (NH3-N) concentrations, mainly by lowering protozoal concentration [11]. Additionally, some phytochemicals have been shown to decrease methane production, either by directly inhibiting methanogenic archaea or by suppressing metabolic pathways involved in methanogenesis [1].

Polyphenols have been linked to improved ruminal fermentation, greater VFA production, and reduced methane emissions [12]. Curcuminoids and piperine have been shown to enhance the performance of dairy cows, antioxidant status, and efficiency [13], while cinnamaldehyde has improved MY efficiency, particularly by reducing DMI [14]. These findings highlight the potential of phytochemicals as feed additives to optimize ruminal fermentation and reduce environmental impacts. Previous studies have reported the potential of phytochemicals to increase MY [15,16] and provide antioxidant, antimicrobial, and anti-inflammatory benefits [17]. Some trials show improved MY and efficiency when combining essential oils and polyphenols [13,18]. However, the results remain inconsistent, as highlighted by a meta-analysis conducted by Khiaosa-ard and Zebeli [10], which found no significant effects from using essential oils alone. Essential oils and phytochemical efficacy are influenced by the interaction between diet composition, additive type, and dosage [1]. Mixed phytochemical compounds may exhibit synergistic or antagonistic effects, with combinations of active ingredients producing varying results [7].

Most research on phytochemicals has focused on total mixed rations in confinement systems, which differ from pasture-based dairy systems in South America, where up to 50% of the diet comes from grazing [19]. Evaluating these additives in mixed systems with pasture inclusion is essential for assessing their potential under local conditions.

We hypothesized that, compared with the control group, phytochemical supplementation improves performance and physiological adaptation to parturition by modulating the ruminal microbial population and fermentation in a manner similar to monensina, thereby improving nutrient utilization. This study aimed to evaluate the effects of a phytochemical mixture and monensin on DMI, body condition score (BCS), MY and milk composition in primiparous and multiparous dairy cows, and ruminal fermentation in primiparous dairy cows during the transition and early lactation period in a mixed grazing system.

## 2. Materials and Methods

The experiment was conducted at the Estación Experimental Dr. Mario A. Cassinoni (EEMAC) of the Facultad de Agronomía (Paysandú, Uruguay, 32° S, 58° W) of the Universidad de la República (UdelaR). Animal procedures were conducted under the approval of the Comisión Honoraria de Experimentación Animal (CEUA) of the Universidad de la República, Montevideo, Uruguay (proceeding number: 020300-000055-22).

### 2.1. Cows, Experimental Design, and Management

Sixty Holstein cows (24 primiparous and 36 multiparous) were used. At the beginning of the study, primiparous cows averaged 581 ± 14.0 kg of body weight (BW) and had a BCS of 3.36 ± 0.27 (scale 1–5; [20]), whereas multiparous cows averaged 655 ± 12 kg of BW and had a BCS of 3.24 ± 0.31. They were assigned to a completely randomized design with randomization restricted to balance for expected calving date and parity. Approximately six months before calving, 9 primiparous cows were fistulated with ruminal cannulas (KEHL, Industria e Comercio LTDA–ME, São Carlos, Brazil). Prior to the beginning of the experimental period, all cows were managed under the same conditions. All animals were kept on the same pasture paddock, separated by parity.

Cows were assigned to one of the three treatments: (1) a control group without additive (CTL), (2) 0.30 g/cow/day monensin (MON; Rumensin, Elanco Animal Health, Greenfield, IN, USA), and (3) 50 g/cow/day of a mix of phytochemical (PHY). The PHY was composed of polyphenols as curcuminoid (up to 280 ppm), trans-cinnamaldehyde (up to 1000 ppm), and piperine (up to 15 ppm), provided by CCPA Group (Janzé, France). The PHY dosage was based on the manufacturer’s recommendation. Both additives were incorporated into the total mixed ration (TMR) and were specific to each group. The final dosages consumed by the cows were 46.4 g/day of PHY and 0.27 g/day of MON. Fifty-one cows completed the study. Data reported are for 16 cows in the CTL group (9 multiparous and 7 primiparous cows), 16 cows in MON (10 multiparous and 6 primiparous cows), and 19 cows in the PHY group (12 multiparous and 7 primiparous cows). Animals were excluded from the experiment due to health problems, and their data were not included in the analysis.

During the experiment, cows were managed in a compost barn, as described by Pons et al. [21]. Thirty days before the expected calving date, cows were moved to the compost barn and housed in pens of four, grouped by category and treatment, where they remained until calving (Figure 1). During this period, cows were fed a TMR formulated to meet their nutritional requirements, following the guidelines of the National Academies of Sciences, Engineering, and Medicine (NASEM; [22]) for dairy cattle.

After calving, and for the first 60 days in milk (DIM), cows grazed daily between 7:00 a.m. and 2:00 p.m. and were supplemented with TMR in the compost barn between afternoon and morning milking (Figure 1). Postpartum, cows were grouped by category and treatment, with each group consisting of either eight primiparous or twelve multiparous cows from the same treatment. This grouping strategy was determined by the experimental conditions and designed to reflect commercial dairy systems in Uruguay. All variables of interest were measured at the individual level; therefore, in this study, the individual cow was considered the experimental unit. To minimize competitive interactions during the grazing period, cows were offered 28.1 kg of DM/cow/day of pasture. While housed in the compost barn, cows were separated by parity to prevent confounding effects during treatment administration. To ensure consistent delivery of the TMR, a daily monitoring protocol was implemented, and each cow was allocated 1 m of feed bunk space to ensure unrestricted access to the assigned diet. The postpartum TMR was formulated to support a MY of 36.0 L/cow/day, assuming an estimated pasture intake of 10.0 kg of dry matter (DM; Table 1). Throughout the experiment, cows grazed Lucerne (*Medicago sativa*, 45.0% part of the time), Tall Fescue (*Lolium arundinaceum*, 25.0% of the time), Chicory (*Cichorium intybus*, 15.0% of the time), or Oat (*Avena sativa*, 15.0% of the time). During grazing, cows were managed separately by treatment in seven-day paddocks, with a target pasture supply of 30.0 kg of DM/cow/day, ensuring pasture intake was not restricted.

Herbage mass was calculated weekly using a double sampling technique adapted from Haydock and Shaw [23] to determine the appropriate paddock sizes. Every week, three replicate sets of three sampling locations were selected within the areas to be grazed. The three locations were chosen according to height to represent a high, medium, and low herbage mass. At each location, sward height was measured using a sward stick, and 30 × 30 cm squares in the same area were cut to ground level with scissors. The cut herbage was collected and weighed to determine DM content, calculate herbage DM mass, and derive a linear regression relating to sward height. The same procedure was used to determine the final mass at the end of the grazing week for better grazing control and to assess that the DMI of pasture was not limited. Pasture availability (mean ± SD) ranged from 1764 ± 100 to 2936 ± 826 kg of DM and height from 15.0 ± 1.28 to 28.8 ± 4.50 cm. The average pasture offered to cows was 28.1 ± 4.71 kg of DM/cow/day.

### 2.2. Sampling and Chemical Analysis

The TMR offered was sampled weekly; subsamples were taken from each feeder immediately after feeding to compose a representative sample of each treatment. Refusals were sampled weekly to estimate DM content, which was used to estimate DMI. Feed samples were analyzed weekly, and the entire period was averaged. Samples of pasture selected by cows were plucked by hand using an adapted technique from Coates and Penning [24]. During the first grazing session, three cows per treatment were followed in the paddock on three specific days of the week (days 1, 3, and 6). Samples of pasture were composed by week and finally by grazing resource. All samples were dried at 60 °C in an air force oven until constant weight and were ground through a Wiley mill (1 mm screen; Thomas Scientific, Swedesboro, NJ, USA) for chemical analysis. Dry matter (AOAC 967.03), ash (AOAC 942.05), ether extract (AOAC 920.39 A), and nitrogen (AOAC 984.13) contents were determined according to the procedure of the AOAC [25]. Organic matter (OM) content was determined as the difference between DM and ash. Crude protein was calculated as nitrogen × 6.25. Neutral detergent fiber (NDF) and acid detergent fiber (ADF) were determined according to Van Soest et al. [26], using an ANKOM200 Fiber Analyzer (ANKOM Technology Corp., Fairport, Macedon, NY, USA) without sodium sulfite, with heat-stable amylase, and expressed on an ash-free basis. Data (Mean ± standard deviation) of the chemical composition of the TMR offered pre- and postpartum, and the average chemical composition of the pasture, are shown in Table 1.

### 2.3. Dry Matter Intake

Dry matter intake of TMR (pre- and postpartum) was calculated as the difference between the feed offered and feed refused or wasted. Prepartum DMI of TMR was measured daily in each pen, with the total intake per pen divided by the number of cows present. Postpartum DMI was measured weekly for each treatment and category. The individual DMI was estimated by dividing the total DMI of the group by the number of cows present at that time. Pasture DMI (kg of DM/cow) was estimated as described by Méndez et al. [27] for 30 and 60 ± 15 DIM by energy balance, calculated as the kg of pasture required to provide the remaining energy to meet the net energy for lactation (NE_L_) requirements of cows that did not come from TMR. The digestible energy (DE) of the pasture was estimated according to Rohweder et al. [28], using the following equation: DE (Mcal/kg) = −0.027 + (0.042 × % organic matter digestibility). Where the organic matter digestibility (OMD) was calculated as OMD (%) = 88.9 − 0.779 × ADF. Metabolizable energy (ME) was estimated with the equation: ME (Mcal/kg) = 1.01 × DE − 0.45 as described by National Research Council (NRC; [29]). The NE_L_ provided by the pasture and the TMR was calculated using the following equations: Pasture NE_L_ (Mcal/kg) = [0.703 × ME of the feed − 0.19] and TMR NE_L_ (Mcal/kg) = 1.909 − 0.017 × ADF [29]. Postpartum TMR energy intake was estimated using weekly DMI data for each group of cows. The NE_L_ requirements of the cows were estimated as the sum of maintenance, activity, and MY requirements, considering the energy contributed or required by the cow for BW and BCS changes, following NASEM [22] equations.

### 2.4. Milk Yield, Composition, and Feed Use-Efficiency

Cows were milked twice daily, at 4:00 a.m. and 4:00 p.m., and their individual MY were recorded daily using automatic milk meters synchronized with DairyPlan software (C21 v5.2, GEA Farm Technologies, Düsseldorf, Germany). Milk composition analysis, including fat, protein, lactose, and urea content, was determined once weekly by mid-infrared spectrophotometry (MilkoScan FT2, Foss, Drachten, The Netherlands). Milk samples were collected during both daily milking, and the daily milk composition was calculated as a weighted average to account for differences in MY between morning and afternoon milkings. Total milk solids (TMSs) were calculated as the sum of lactose, protein, and fat yield (kg/day). Feed use efficiency was determined as the ratio of MY to DMI. Energy-corrected milk (ECM) was calculated according to the guidelines of the NRC [29] as follows: NE_L_ (Mcal/d) = [0.0929 × kg of fat + 0.0547 × kg of protein + 0.0395 × kg of lactose] × 100.

### 2.5. Ruminal Fermentation Characteristics and Protozoa Concentration

Ruminal fluid samples were collected from fistulated cows (n = 3 per treatment) at 7 days prepartum, and at 30 and 60 DIM (30-day and 60-day periods). During the two postpartum sampling periods, the cows grazed exclusively on lucerne. The samples were collected from different portions of the rumen using a manual extraction device equipped with a screening tube in the last part. Samples were collected over two consecutive days at 0, 4, 8, 16, and 24 h relative to morning feeding to determine pH, NH3-N, and VFA. Samples were preserved according to the method described by Mattiauda et al. [30]. Immediately after collection, pH was measured using a portable pH-meter (Oakton, Eutech Instruments, Melaka, Malaysia). Volatile fatty acid and NH3-N samples were centrifuged at 10,000× *g* for 10 min at 4 °C and stored at −20 °C until analysis. Volatile fatty acid concentrations were determined by gas chromatography (GC; Agilent 8860, Agilent Technologies, Santa Clara, CA, USA) following filtration of rumen fluid samples through a 0.45 μm membrane filter. The GC system was equipped with an automatic injector, a flame ionization detector (FID), and an HP-INNOWax capillary column (30 m × 0.53 mm, 1 μm film thickness). Nitrogen was used as the carrier gas, and the detector temperature was maintained at 300 °C. Quantification of individual VFAs (acetic, propionic, butyric, isobutyric, valeric, and isovaleric acids) was performed using external calibration curves constructed from pure analytical standards (acetic acid: Supelco, Merck, Darmstadt, Germany; other acids: Sigma-Aldrich, Schnelldorf, Germany) prepared at known concentrations. Calibration curves were generated under the same analytical conditions as the samples and showed high linearity (R^2^ > 0.99). Ammonia concentration was determined using spectrophotometry according to the method of Chaney and Marbach [31]. Protozoa concentration was determined using the technique described by Dehority [32], adapted by D’Agosto and Carneiro [33] to use Lugol (5%). Protozoa were counted using a double-reticle Neubauer chamber (Neubauer Improved, Marienfeld Superior, Lauda-Königshofen, Germany), and samples were examined under an optical microscope (Labomed (CxL), Los Angeles, CA, USA) at 10× magnification

### 2.6. Statistical Analysis

The data were analyzed using the SAS software (version 9.4; SAS OnDemand for Academics; SAS Institute Inc., Cary, NC, USA). A completely randomized design was applied using PROC GLIMMIX, and data collected at different time intervals were analyzed as repeated measures. Data on DMI, BCS, MY, and composition were analyzed separately by parity. Prepartum DMI was analyzed with treatment included as a fixed effect and pen nested within treatment as a random effect; feed supply was included as a covariate. For postpartum DMI the model included the fixed effect of treatment, DIM (30 and 60 days), and their interactions, with BW as a covariate and the cow as a random effect. The statistical model for BCS included the fixed effects of treatment, week, and their interaction, with BCS at the beginning of the experiment as a covariate. Milk yield and composition were analyzed using a model that included treatment as a fixed effect and cow as a random effect, with the BW at calving, DIM, and DIM^2^ as covariates. Ruminal fermentation data (VFA, NH3-N, pH, and protozoa) were analyzed separately for prepartum and postpartum periods. The model included treatment, hour (prepartum), and hour nested in period (postpartum) as fixed effects and their interactions, and cow as a random effect. Post hoc analyses were carried out through the Tukey–Kramer test. Significant differences were considered when *p* ≤ 0.05 and as a tendency at *p* ≤ 0.10.

## 3. Results

### 3.1. Dry Matter Intake and Body Condition Score

Before calving, PHY primiparous cows did not differ in DMI from CTL cows but had greater (*p* = 0.01) DMI than MON primiparous cows (2.06 and 1.98 ± 0.03 vs. 1.88 ± 0.02% of BW, respectively). The additives did not affect (*p* = 0.574) prepartum DMI in multiparous cows (CTL: 2.21, MON: 2.17, PHY 2.14 ± 0.07% of BW). None of the feed additives affected postpartum DMI. However, DMI was lower (*p* < 0.01) at 30 DIM compared to 60 DIM for both multiparous (3.70 vs. 3.90 ± 0.06% of BW, respectively) and primiparous cows (3.48 vs. 3.87 ± 0.12% of BW, respectively). Treatments tended to affect BCS in primiparous cows (*p* = 0.10). While BCS did not differ between PHY cows and those in the MON and CTL groups (3.13, 3.21, and 3.09 ± 0.02, respectively), MON cows tended to have a higher BCS compared to the CTL group (*p* = 0.08). In multiparous cows, no differences were observed between groups (3.00 ± 0.02). The body condition score was affected by the week (*p* < 0.01), and the lowest BCS was reached six weeks postpartum in both primiparous (2.86 ± 0.04) and multiparous cows (2.71 ± 0.05).

### 3.2. Milk Yield and Composition

Supplementation with additives increased MY (*p* = 0.01; 7.93%) and milk lactose *(p* = 0.04; 2.36%) concentration in primiparous cows (Table 2). Multiparous PHY cows did not differ in MY from CTL cows but tended to have lower MY than MON cows (*p* = 0.07). Neither fat, protein, TMS, ECM, nor feed efficiency (kg of milk/kg of DMI) was affected by additives (Table 3).

### 3.3. Rumen Fermentation Characteristics and Protozoa

No effect of additives was found on ruminal prepartum pH, NH3-N, total VFA, or in the proportion of acetate (C2), propionate (C3), and butyrate (C4). The percentage of branched-chain volatile fatty acids (BCVFAs) in PHY cows did not differ from MON cows and was higher (*p* < 0.05) than in CTL cows, whereas MON cows did not differ from CTL cows. There was no difference in prepartum protozoa concentrations between MON and PHY cows, while MON cows had lower values (*p* < 0.01) than CTL cows. No interaction between treatment and hour was observed for any prepartum parameters (Table 4).

Significant interactions for ruminal postpartum pH were observed between treatment and period, as well as between treatment and hour (Table 5). Rumen pH values were higher (*p* < 0.01) during the 30-day period compared to the 60-day period (6.67 ± 0.02 vs. 6.34 ± 0.03, respectively). Rumen pH remained stable across all treatments during the first 8 h, when the cows were grazing, and then decreased, reaching its lowest values at 16 h after feeding (Figure 2A,B). In a 30-day period after calving, ruminal pH in PHY and MON cows was lower (*p* = 0.01) at 16 h than at the rest measurement hours. However, in CTL cows, pH at 16 h differed only from 4 and 8 h (Figure 2A). In the 60-day period, pH at 16 h differed (*p* < 0.01) only in PHY cows from all other measurement hours.

Postpartum total VFA was affected by the sampling hour and by the interaction between treatment and period. Maximum concentrations of VFA were observed at 16 h in all groups (Figure 2C,D). Cows supplemented with additives showed lower (*p* < 0.05) total VFA concentration during the 30-day period (104 and 104 vs. 130 ± 4.85 mmol/L for PHY and MON vs. CTL, respectively). However, during the 60-day period, total VFA did not differ between groups and followed similar patterns (114 and 115 vs. 111 ± 1.71 mmol/L for PHY and MON vs. CTL, respectively; Figure 2D).

Acetate, C3, C2:C3, and ketogenic-to-glucogenic ratio (C2 + C4:C3) concentration were affected by the interaction between treatment and period (Table 5). Acetate and C3 concentrations and C2:C3 and C2 + C4:C3 ratios were not different between groups during the 30-day period (Figure 3A). However, in the 60-day period, the proportion of C2 was higher (*p* = 0.05) in CTL cows compared to MON and PHY cows (62.5 vs. 60.2 and 59.9 ± 0.85%, respectively; Figure 3B). Additionally, C3 concentration in the 60-day period was higher (*p* = 0.01) in PHY cows compared to CTL cows (27.3 vs. 23.4 ± 0.78%), while MON cows were intermediate (26.4 ± 0.78%; Figure 3B). This resulted in a lower C2:C3 ratio (*p* = 0.05) in PHY cows compared to CTL cows (2.27 vs. 2.72 ± 0.10, respectively). The C2 + C4:C3 ratio was lower (*p* < 0.02) in PHY than in CTL cows (2.66 vs. 3.26 ± 0.12, respectively), and MON cows were intermediate (2.76 ± 0.12).

Butyrate concentration was affected by the treatment and the period. It was lower (*p* < 0.01) in PHY cows compared to MON and CTL cows, and was lower (*p* = 0.01) in the 30-day period compared to the 60-day period (11.01 vs. 11.95 ± 0.23%, respectively). Additionally, BCVFA concentration was lower (*p* = 0.01) in cows supplemented with additives (Table 5).

The concentration of NH3-N was affected by treatment, period, hour, and interactions (Table 5, Figure 2E,F). In general, PHY cows exhibited lower (*p* < 0.01) NH3-N concentrations than MON and CTL. Ammonia concentrations were lower in the 30-day period compared to the 60-day period (69.4 vs. 88.6 ± 2.11 ppm, respectively). In the 30-day period, NH3-N was lower in PHY and MON compared to CTL cows (61.1 and 60.3 vs. 86.9 ± 3.53 ppm, respectively). However, in the 60-day period, PHY cows had lower NH3-N concentration than MON and CTL cows (70.4 vs. 103 and 91.8 ± 3.77 ppm, respectively). The NH3-N concentration fluctuated throughout the day, with the highest concentration observed at 8 h (Figure 2). At the 30-day period, the NH3-N concentration at 4 h was higher in CTL cows than in MON (*p* = 0.01) and PHY (*p* < 0.01) cows, with no difference observed between MON and PHY (Figure 2E). Conversely, at the 60-day period, PHY cows exhibited lower NH3-N concentrations at 4 and 8 h compared to CTL (*p* = 0.01 and *p* = 0.03, respectively) and MON (*p* < 0.01 and *p* = 0.01, respectively) cows (Figure 2F).

Both additives reduced (*p* < 0.01) protozoa postpartum concentration compared to CTL cows, and there was no difference between them (Table 5). Period affected protozoa concentration; they were greater in the 60-day period than in the 30-day period (103 vs. 59.5 cells × 10^4^/^mL^, respectively).

## 4. Discussion

To the best of our knowledge, this is the first report to compare the effects of a phytochemical mixture versus monensin and no additives applied during the transition period and early lactation on DMI, BCS, MY, milk composition, ruminal fermentation parameters, and protozoa concentration in dairy cows under a mixed grazing system. Data are consistent regarding the effect of the PHY in reducing protozoa and shifting to a more glucogenic ruminal environment while decreasing NH3-N ruminal concentrations. However, the benefits of the additive were observed only in primiparous cows.

### 4.1. Productive Performance

The use of PHY and MON improved productive performance in primiparous cows, but not in multiparous cows, highlighting the variability in responses to additives [10,34,35]. It is well established that primiparous cows face greater physiological and metabolic challenges compared to multiparous cows [36,37,38]. In a parallel study, Comesaña et al. ([39]; unpublished data) observed higher concentrations of key metabolic indicators (beta-hydroxybutyrate and non-esterified fatty acids) in primiparous cows compared to multiparous cows. Additives may have helped to lift these challenges in primiparous cows. Although no response in MY was observed in multiparous cows, they appeared to manage the transition period effectively. This was evidenced by their BCS and MY, which suggest successful adaptation to the demands of lactation.

Primiparous cows supplemented with MON managed nutrients more efficiently during the transition, as indicated by their lower prepartum DMI and improved postpartum BCS, consistent with previous studies [40,41,42]. The reduction in DMI could be associated with ruminal filling effects due to lower prepartum feed degradability [43]. This observed decrease in protozoal concentration in this study likely contributed to reducing digestibility and DMI, consistent with Newbold et al. [44]. A parallel study (Dorao et al., [45]; unpublished data) reported that PHY and MON supplementation reduced OMD in the prepartum diet.

Milk yield is primarily influenced by DM and nutrient intake [46]. In our study, there was no discernible effect of additives on postpartum DMI in both categories. The increase in MY (7.93%) and lactose concentration (2.36%) resulting from the use of both additives in primiparous cows suggests greater glucose availability for their synthesis. Glucose is the most important energy substrate for promoting metabolism and increasing MY, particularly in grazing dairy cows, since it is derived almost exclusively from hepatic gluconeogenesis [47]. This greater glucose availability can be attributed to an increased supply of gluconeogenic precursors [48]. In general, studies of monensin [4,41] or phytochemicals [10,13,18] that report improvements in MY attribute these benefits to increased fermentation efficiency in the rumen. This increased efficiency is a result of higher glucogenic precursors, such as C3, better absorption of gluconeogenic amino acids, and reduced energy loss [18].

The observed increase in MY agrees with the ranges reported for monensin [42,49,50] and phytochemicals [18,51]. However, the search for similar responses regarding the increase in lactose concentration has yielded divergent results. Some studies have reported an increase in lactose concentration with the use of phytochemicals [35,52].

### 4.2. Rumen Fermentation Profile

During the prepartum period, PHY increased BCVFA, while both additives reduced protozoa concentration. In contrast, the postpartum period showed treatment-specific responses influenced by sampling time (30-day or 60-day period), highlighting the variability in responses to additive use at different physiological stages and diets. These results are consistent with studies showing that animal diet quality and physiological status influence their responses to additives [34,35,53].

The increase in BCVFA during the prepartum period could suggest reduced utilization by cellulolytic bacteria, potentially linked to the observed reduction in OMD. These findings align with those of Tomkins et al. [54], who observed similar BCVFA increases with phytochemical supplementation (CRINA) in medium-quality hay-based diets like our pre-partum diet. The postpartum reduction in BCVFA likely reflects decreased protein degradation, as reduced ruminal NH3-N levels indicate. Studies using different phytochemicals in higher-quality diets reported no changes in BCVFA concentration [51,55,56].

The reductions in protozoa observed with PHY and MON are attributed to their lipophilic properties, which may alter cell membrane permeability [1,2]. Reductions in protozoan populations have been reported with saponins and tannins [11,12], the inclusion of 200 g/d of peppermint in steer diets [57], and with essential oils at different doses [34,58]. Conversely, McIntosh et al. [59] and Benchaar et al. [60] reported no effect on protozoan populations with essential oil supplementation (CRINA; 2 g/day). Interestingly, Khiaosa-ard and Zebeli [10] found that high doses (0.5 g/kg DM) of essential oils reduced protozoa, whereas lower doses tended to increase them. These findings are consistent with the present study, where higher doses of PHY demonstrated an inhibitory effect on protozoa.

The pH, total VFA concentrations, and their patterns were consistent with previous findings in supplemented grazing cows [61] and are typical of the feeding strategies employed. The decrease in pH at the 60-day period could be associated with a greater supply of fermentable OM and increased VFA production, likely related to a greater DMI [62].

Volatile fatty acid concentrations peaked at 16 h across all groups, coinciding with peak feed utilization in the afternoon (Figure 2A,B). At the 30-day postpartum period, PHY and MON cows exhibited more stable VFA patterns between 0 and 8 h compared to CTL cows, which showed a pronounced drop. By the 60-day period, VFA patterns became consistent across treatments, likely due to greater stability in DMI and adaptation to the postpartum diet.

The lower total VFA concentration observed at the 30-day period postpartum with additives suggests a temporary reduction in dietary fermentability, consistent with the findings of Kholif et al. [35]. In the 60-day period, these effects disappeared, according to the findings that OMD was unaffected by MON or PHY supplementation [45]. Similarly, Benchaar et al. [53] observed variability in in vitro experiments, where a blend of essential oil compounds increased total VFA concentrations in a forage-based diet but reduced them in a concentrate-based diet.

In a 60-day period, the additive supplementation reduced C2 production. Additionally, PHY lowered C4 concentrations as reported by Mezzetti et al. [63] and increased C3 production as reported by Kholif et al. [35]. These shifts can be attributed to the inhibition of Gram-positive bacteria [64,65]. As a result of these changes, PHY supplementation reduced the C2 + C4:C3 ratio by 18.4% compared to CTL cows. This shift suggests more glucogenic fermentation and, consequently, a more efficient ruminal profile.

Studies on phytochemicals and monensin have reported increased C3 concentrations ranging from 12% to 43% in cows fed TMR [13,35,66]. However, Flores et al. [67] found no changes in ruminal C3 concentration or the C2 + C4:C3 ratio when grazing cows were supplemented with low doses of a phytochemical mixture (0.20–0.60 g/cow/day).

Both additives reduced NH3-N levels, but PHY demonstrated greater consistency across postpartum days. This reduction could be attributed to decreased deamination and degradation of peptides and amino acids due to reduced activity of hyper-ammonia-producing bacteria [7,58] and protozoa [11]. Also, unlike protozoa, bacteria that replace protozoa can assimilate NH3-N [2]. Lower ruminal NH3-N concentrations might indicate greater escape of undegraded protein from the rumen, as Hayes et al. [68] indicated. This suggests potentially enhancing amino acid availability for absorption in the small intestine.

In our study, PHY supplementation improved rumen stability compared with MON, likely due to differences in ruminal microbiota adaptation [69]. Although the PHY mixture also exerts antimicrobial activity, its effects may involve multiple mechanisms, including disruption of microbial membranes, enzyme inhibition, and interference with cell signaling [7,11]. These diverse actions could potentially limit microbial adaptation during the evaluation period. However, with prolonged exposure, microbial adaptation to these mechanisms may still occur, raising concerns similar to those associated with monensin and highlighting the need for further investigation. The modulating protein degradation without inhibiting bacterial growth enhances nitrogen utilization efficiency, potentially saving energy by reducing NH3-N absorption, urea conversion, and excretion [70]. This positions PHY as a promising additive to maintain animal performance while reducing environmental impacts.

The greater glucogenic profile, coupled with reduced NH3-N concentrations and potentially a greater amino acid absorption, underscores the potential of PHY to improve metabolic efficiency, MY, and energy efficiency in lactating cows. In our study, these effects were particularly evident when DMI was stabilized. Zhang et al. [71] linked glucogenic fermentation with improved energy efficiency, while Larsen and Kristensen [48] and Larsen et al. [72] associated it with enhanced amino acid absorption, contributing to greater overall efficiency in dairy production. Factors such as the dosage of additives, diet quality, and shifts in the ruminal microbiota likely contribute to more efficient fermentation. This warrants further investigation to better understand the effects of these additives on ruminal ecology and their implications for dairy cattle nutrition and productivity.

While the use of the individual animal as the experimental unit may be considered a limitation in dairy studies where animals share a common environment (e.g., a pen), this approach may be valid when the effects of the group or shared environment are adequately controlled (as previously described). Furthermore, it offers greater sensitivity for detecting biologically significant individual variations relevant to the study objectives. However, these results should be interpreted with caution, as the animal was used as the experimental unit despite treatments being applied at the group level. Therefore, further studies are warranted to confirm our findings.

## 5. Conclusions

In dairy cows fed fresh pasture and TMR, supplementation with PHY and MON improved lactation performance in primiparous cows. This benefit was associated with changes in ruminal fermentation, including a shift towards more glucogenic pathways, a reduction in protozoal concentration, and lower NH3-N concentrations, potentially enhancing the absorption of glucogenic amino acids in the small intestine. These changes likely contributed to more efficient energy and nitrogen utilization, with the potential to reduce environmental externalities. Furthermore, PHY-supplemented cows demonstrated greater stability over time.

## Figures and Tables

**Figure 1 animals-15-02172-f001:**
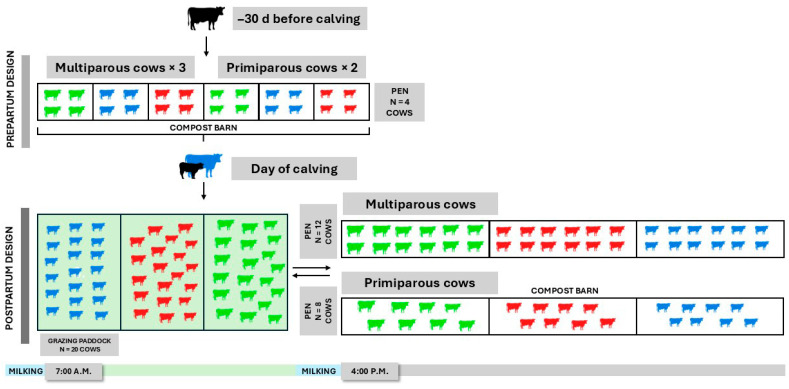
Schematic representation of the experimental design during prepartum (−30 days [d]) and postpartum period. Green, red, and blue cows represent the control, monensin (MON), and phytochemicals (PHY) treatments, respectively.

**Figure 2 animals-15-02172-f002:**
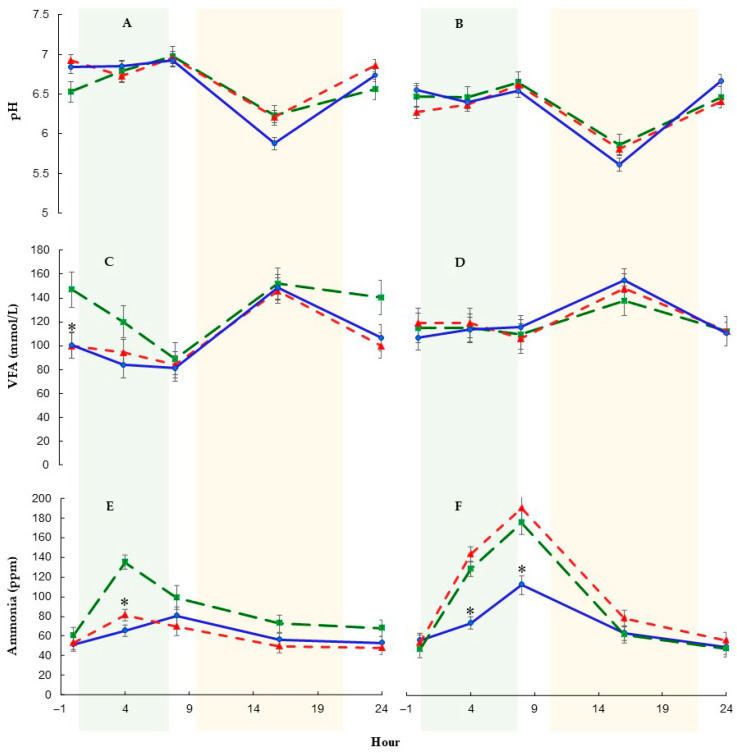
Ruminal pH (**A**,**B**), total volatile fatty acids (VFAs) concentration (**C**,**D**) and ammonia (**E**,**F**) for CTL (green), MON (red) and PHY (blue) at 30-day period (**A**,**C**,**E**) and 60-day period postpartum (**B**,**D**,**F**). The green shaded area represents the hours that the cows were grazing. The yellow shaded area represents the hours that the cows were in the composted barn with total mixed ration supplementation. * *p* < 0.05.

**Figure 3 animals-15-02172-f003:**
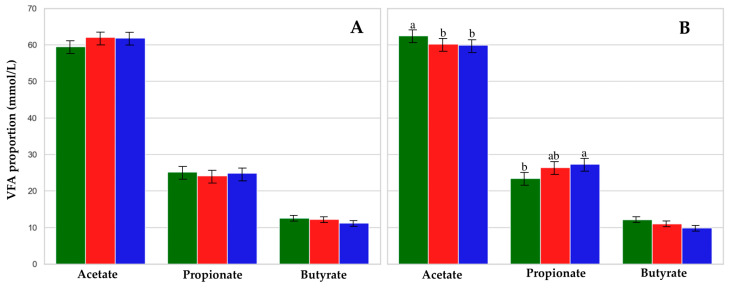
Proportion of volatile fatty acids (VFAs) for CTL (green), MON (red), and PHY (blue) at 30-day (**A**) and 60-day period postpartum (**B**). Letters indicate differences (*p* < 0.05) between treatments within each period.

**Table 1 animals-15-02172-t001:** Ingredient composition (% of dry matter [DM]) and chemical composition (mean ± standard deviation) of the total mixed ration (TMR; basal diet) and pasture.

Ingredient (% of DM)	TMR Prepartum	TMR Postpartum	Pasture ^5^
Corn Silage	38.8	30.6	-
Barley Straw	31.4	-	-
Alfalfa Hay	-	13.6	-
Soybean Meal	16.2	13.6	-
Grain Corn	4.37	26.4	-
Wheat Bran	4.42	-	-
Barley grain	-	7.65	-
Calcium carbonate	1.36	1.69	-
Sodium chloride	-	0.68	-
Bicarbonate	-	0.66	-
Magnesium oxide	-	0.66	-
Monocalcium phosphate	-	0.66	-
Sulfur	-	1.74	-
Semitin wheat	-	1.74	-
Mineral and Vitamin supplement ^1^	3.45	0.32	-
Chemical Composition			
Organic Matter	90.6 ± 1.34	91.3 ± 0.44	88.1 ± 2.1
Crude Protein	10.8 ± 1.48	12.8 ± 0.96	17.4 ± 0.6
NDF ^2^	44.2 ± 2.51	26.8 ± 3.44	36.8 ± 7.0
ADF ^3^	25.9 ± 1.02	14.0 ± 1.50	22.4 ± 2.6
Digestible Energy (Mcal/Kg of DM)	2.86 ± 0.01	3.07 ± 0.01	2.97 ± 0.09
NE_L_ ^4^ (Mcal/Kg of DM)	1.47 ± 0.01	1.67 ± 0.01	1.61 ± 0.06

^1^ Contained prepartum: −9831 meq/kg, 6% Mg, 0.20% K, 2.0% Na, 16% Cl, 10% S, 10% Ca, 250 mg/Kg Cu, 8.0 mg/Kg Se, 1200 mg/Kg Zn, 9.0 mg/Kg I, 2.4 mg/Kg Co, 210 UI/Kg Vitamin A, 90,000 UI/Kg Vitamin D, 2400 UI/Kg Vitamin E. Postpartum: 20,000 mg/Kg Zn, 4000 mg/Kg Cu, 120 mg/Kg Se, 160 mg/Kg I, 40 mg/Kg Co, 120,000 UI/Kg Vitamin A, 240,000 UI/Kg Vitamin D3, 8400 UI/Kg Vitamin E; ^2^ NDF = Neutral detergent fiber; ^3^ ADF = Acid detergent fiber; ^4^ NE_L_ = Net Energy of lactation of TMR (Mcal/kg) calculated as follow: 1.909 − (0.017 × ADF) and NE_L_ of Pasture (Mcal/kg) = [0.703 × Mcal of metabolizable energy − 0.19]; ^5^ Pasture composition represents the average of the feed selected by the cows during the experimental period.

**Table 2 animals-15-02172-t002:** Milk yield and milk composition in primiparous grazing cows fed a total mixed ration without supplementation (CTL) or supplemented with monensin (MON) or phytochemicals (PHYs).

	Treatment				*p*-Value			
Item	CTL	MON	PHY	SEM	T ^5^	BW ^6^	DIM ^7^	DIM^^2^
Production								
Milk yield (kg/day)	26.5 ^b^	28.5 ^a^	28.7 ^a^	0.52	0.01	<0.01	<0.01	<0.01
ECM ^1^ (Mcal/day)	19.5	21.0	20.2	0.86	0.50	0.47	0.21	0.41
TMS ^2^ (kg/day)	3.24	3.49	3.39	0.20	0.44	0.50	0.06	0.19
Feed efficiency ^3^	1.44	1.45	1.51	0.04	0.54	-	-	-
Composition (%)								
Fat	3.71	3.73	3.59	0.15	0.53	0.49	<0.01	<0.01
Protein	3.38	3.46	3.31	0.05	0.22	0.97	<0.01	<0.01
Lactose	4.88 ^b^	5.00 ^a^	4.99 ^a^	0.05	0.04	0.55	<0.01	<0.01
MUN ^4^ (mg/dL)	14.2	14.5	13.0	0.99	0.54	0.69	0.14	0.29
Yield (kg/d)								
Fat	0.99	1.05	1.02	0.05	0.76	0.28	0.38	0.55
Protein	0.91	0.98	0.94	0.03	0.35	0.45	0.30	0.52
Lactose	1.30	1.43	1.42	0.04	0.11	0.49	<0.01	<0.01

^a,b^ Means within a row with different superscripts differ (*p* ≤ 0.05). ^1^ ECM (Energy-Corrected Milk) calculated as ECM (Mcal/d) = [0.0929 × kg fat + 0.0547 × kg protein + 0.0395 × kg lactose] × 100 [27]. ^2^ TMS = Total milk solids calculated as kg fat + kg protein + kg lactose (kg/d). ^3^ Feed efficiency = Milk (kg/day)/Dry matter intake (kg/day). ^4^ MUN = Milk Urea Nitrogen. ^5^ T = Treatment. ^6^ BW = Body Weight. ^7^ DIM = Days in milk.

**Table 3 animals-15-02172-t003:** Milk yield and milk composition in multiparous grazing cows fed a total mixed ration without supplementation (CTL) or supplemented with monensin (MON) or phytochemicals (PHYs).

	Treatment				*p*-Value			
Item	CTL	MON	PHY	SEM	T ^5^	BW ^6^	DIM ^7^	DIM^^2^
Production								
Milk yield (kg/day)	36.9	37.7	36.0	0.48	0.07	<0.01	<0.01	<0.01
ECM ^1^ (Mcal/day)	25.9	26.3	25.5	0.55	0.50	<0.01	0.18	0.26
TMS ^2^ (kg/day)	4.31	4.41	4.28	0.13	0.55	<0.01	<0.05	0.10
Feed efficiency ^3^	1.56	1.59	1.61	0.03	0.56	-	-	-
Composition (%)								
Fat	3.68	3.54	3.44	0.09	0.19	0.61	<0.01	<0.01
Protein	3.41	3.31	3.33	0.04	0.29	0.78	<0.01	<0.01
Lactose	4.83	4.88	4.88	0.03	0.56	0.93	<0.01	<0.01
MUN ^4^ (mg/dL)	13.4	15.4	13.6	0.62	0.08	0.69	0.13	0.47
Yield (kg/d)								
Fat	1.30	1.32	1.25	0.03	0.21	<0.01	0.52	0.50
Protein	1.23	1.24	1.22	0.03	0.83	<0.01	0.23	0.36
Lactose	1.73	1.84	1.78	0.04	0.25	<0.01	<0.01	<0.01

^1^ ECM (Energy-Corrected Milk) calculated as ECM (Mcal/d) = [0.0929 × kg fat + 0.0547 × kg protein + 0.0395 × kg lactose] × 100 [27]. ^2^ TMS = Total milk solids calculated as kg fat + kg protein + kg lactose (kg/d). ^3^ Feed efficiency = Milk (kg/day)/Dry matter intake (kg/day). ^4^ MUN = Milk Urea Nitrogen. ^5^ T = Treatment. ^6^ BW = Body Weight. ^7^ DIM = Days in milk.

**Table 4 animals-15-02172-t004:** Effect of dietary treatment (T) on ruminal fermentation characteristics and protozoa concentration 7 days prepartum in cows fed a total mixed ration without supplementation (CTL) or supplemented with monensin (MON) or phytochemicals (PHYs).

	Treatment					*p*-Value	
Item	CTL	MON	PHY	SEM	T	Hour	T × Hour
Ruminal pH	6.50	6.53	6.66	0.06	0.18	<0.01	0.83
Ammonia (ppm)	101	101	77.3	9.70	0.14	0.20	0.36
Total VFA ^1^	91.5	94.8	103	7.95	0.57	0.33	0.99
Acetate (%)	62.9	64.7	61.1	1.56	0.24	0.12	0.96
Propionate (%)	23.2	23.2	22.2	0.89	0.63	0.40	0.77
Butyrate (%)	11.1	11.0	11.6	0.97	0.88	0.07	0.98
Valerate (%)	1.18 ^a^	0.96 ^b^	1.16 ^ab^	0.07	0.05	0.02	0.99
BCVFA ^2^ (%)	1.22 ^b^	1.30 ^ab^	1.78 ^a^	0.15	0.04	0.10	0.99
C2:C3 ^3^	2.72	2.91	2.73	0.16	0.64	0.32	0.99
C2 + C4:C3 ^4^	3.22	3.26	3.48	0.17	0.55	0.46	0.90
Protozoa ^5^	12.3 ^a^	6.72 ^b^	8.23 ^ab^	1.35	0.01	0.01	0.88

^a,b^ Means within a row with different superscripts differ (*p* ≤ 0.05). ^1^ Total VFA = Total volatile fatty acids concentration (mmol/L). ^2^ BCVFA = Branched chain volatile fatty acids. ^3^ C2:C3 = Acetate/propionate ratio (C2:C3). ^4^ C2 + C4:C3 = Ketogenic/Glucogenic ratio. ^5^ Protozoa = Protozoa concentration (cells/mL ×10^4^).

**Table 5 animals-15-02172-t005:** Effect of dietary treatment (T) on ruminal fermentation characteristics and protozoa concentration postpartum (in period 30 and 60 days) in grazing cows fed a total mixed ration without supplementation (CTL) or supplemented with monensin (MON) or phytochemicals (PHYs).

	Treatment				*p*-Value				
Item	CTL	MON	PHY	SEM	T	Hour	Per ^6^	T × Per	T × Hour
Ruminal pH	6.49	6.51	6.49	0.03	0.90	<0.01	<0.01	0.04	0.01
Ammonia (ppm)	89.4 ^a^	82.0 ^a^	65.8 ^b^	2.59	<0.01	<0.01	<0.01	<0.01	<0.01
Total VFA ^1^	121	110	109	3.85	0.09	<0.01	0.95	0.02	0.91
Acetate (%)	62.0	61.1	60.9	0.54	0.95	<0.01	0.70	0.01	0.99
Propionate (%)	24.3	25.2	26.1	0.55	0.08	0.20	0.14	0.02	0.99
Butyrate (%)	12.3 ^a^	11.6 ^a^	10.5 ^b^	0.28	0.01	<0.01	0.01	0.50	0.61
Valerate (%)	1.56 ^a^	1.18 ^b^	1.42 ^a^	0.07	0.01	0.24	0.14	0.05	0.69
BCVFA ^2^ (%)	1.15 ^a^	0.93 ^b^	0.93 ^b^	0.04	0.01	<0.01	0.06	0.36	0.93
C2:C3 ^3^	2.57	2.45	2.38	0.07	0.24	0.04	0.48	0.02	0.99
C2 + C4:C3 ^4^	3.06	2.91	2.78	0.09	0.09	0.40	0.61	0.03	0.98
Protozoa ^5^	11.6 ^a^	7.12 ^b^	4.03 ^b^	1.10	<0.01	0.62	0.02	0.17	0.99

^a,b^ Means within a row with different superscripts differ (*p* ≤ 0.05). ^1^ Total VFA = Total volatile fatty acids concentration (mmol/L). ^2^ BCVFA = Branched chain volatile fatty acids. ^3^ C2:C3 = Acetate/propionate ratio (C2:C3). ^4^ C2 + C4:C3 = Ketogenic/Glucogenic ratio. ^5^ Protozoa = Protozoa concentration (cells/mL ×10^4^). ^6^ Per = Period of ruminal sampling (30 and 60 days postpartum).

## Data Availability

The data presented in this study are available upon reasonable request from the corresponding author due to considerations related to project funding agreements.

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
