# Peer review of "Phytochemicals and Monensin in Dairy Cows: Impact on Productive Performance and Ruminal Fermentation Profile"

_animals, 2025, doi:10.3390/ani15152172_

Round 1
Reviewer 1 Report
Comments and Suggestions for Authors
Review of the Manuscript: Phytochemicals and Monensin in Dairy Cows: Impact on Productive Performance and Ruminal Fermentation Profile
Simple Summary
The current summary only provides an introduction. Could you include key findings to give readers a clearer overview of the results?
Abstract
Please add a brief description of the statistical analysis used. Additionally, the hypothesis should be stated more explicitly. How do you expect different phytochemicals to influence pasture conditions?
Materials and Methods
Were the animals pregnant heifers or primiparous cows at the trial's start?
How was the phytochemical (PHY) dosage determined?
Could the 15% animal loss during the study impact treatment outcomes?
How was dry matter intake (DMI) assessed during the pre-partum period in collective barns? What measures were taken to account for non-study cows?
Table 1. How was the variability in diet composition evaluated?
Why were primiparous and multiparous cows analyzed separately? Could parity be included as a variable in the statistical model?
Is a sample size of only three cows per treatment sufficient to assess treatment effects reliably?
Results
Data on DMI and body condition score (BCS) were not presented in the tables. Could these be included?
Why were ruminal fermentation evaluations split at 7 days in milk (DIM) but combined for 30 and 60 DIM?
Was the feed intake of rumen-cannulated cows monitored during fermentation assessments?
Author Response
Comments and Suggestions for Authors – Reply to the Review Report
- Summary
Thank you very much for taking the time to review our manuscript. We carefully considered each of your comments. Please find our detailed responses below, with the corresponding corrections highlighted in yellow in the re-submitted file.
Review of the Manuscript: Phytochemicals and Monensin in Dairy Cows: Impact on Productive Performance and Ruminal Fermentation Profile
Simple Summary
Comment 1: The current summary only provides an introduction. Could you include key findings to give readers a clearer overview of the results?
Response 1: Thank you for your recommendation. We have incorporated the key findings into the summary section, after line 7 of the manuscript. The added text is highlighted in yellow for your convenience.
Abstract
Comment 2: Please add a brief description of the statistical analysis used. Additionally, the hypothesis should be stated more explicitly. How do you expect different phytochemicals to influence pasture conditions??
Response 2: Thank you for your helpful suggestion. Regarding the inclusion of statistical analysis in the abstract, we carefully considered your comment alongside the Animals journal guidelines, which specify that the abstract should be a single paragraph without headings and should include background, methods, key findings, and conclusions. Due to the strict word limit (200 words), including statistical details would require omitting essential biological content. Therefore, we choose to include a concise description of the statistical methods in the main text, where full methodological details are more appropriately presented. We hope this approach aligns with both the journal's format and your expectations.
Additionally, due to the word constraint, we stated the hypothesis in the Introduction section (lines 78 to 82), where we considered it more fitting to provide a clearer rationale for the study. We trust this also aligns with your recommendation.
Materials and Methods
Comment 3: Were the animals pregnant heifers or primiparous cows at the trial’s start?
Response 3: Thank you for the observation. We have added information in lines 96 to 98 to clarify this point. Prior to the start of the experiment, both primiparous and multiparous cows within each treatment group were managed under the same conditions. All animals were maintained on pasture, ensuring similar nutritional and environmental backgrounds before the trial began.
Comment 4: How was the phytochemical (PHY) dosage determined?
Response 4: Thank you for your question. This is important information that we should have included in the manuscript. We have added this information to lines 103-104 of the manuscript. The PHY dosage was determined according to the manufacturer's recommendation (CCPA Group), based on previous internal data and trials.
Comment 5: Could the 15% animal loss during the study impact treatment outcomes?
Response 5: Thanks for your question. At the beginning of the experiment, the number of animals per group accounted for potential exclusions of animals due to health reasons, given the inherent challenges of conducting studies during the transition period. Although the loss of approximately 15% of the animals could affect the results, we think that the final number of animals was sufficient to support the validity of the treatment comparisons and the conclusions drawn.
Comment 6: How was dry matter intake (DMI) assessed during the pre-partum period in collective barns? What measures were taken to account for non-study cows?
Response 6: We appreciate the opportunity to clarify this point. As described in lines 164–169 of the manuscript, prepartum DMI of the TMR was recorded daily at the pen level, and total intake was divided by the number of cows present in the pen to estimate average intake per cow. Only study animals were housed in those pens during the experimental period; therefore, no adjustments for non-study cows were necessary. Postpartum DMI was recorded weekly for each treatment and category, and individual DMI was similarly estimated by dividing total group intake by the number of cows present. If you need more information regarding this point, we can work on it.
Comment 7: Table 1. How was the variability in diet composition evaluated?
Response 7: Thank you for your question. To evaluate the variability in diet composition, we conducted weekly analysis (lines 146-147) of the feed samples collected throughout the experiment. The values presented in Table 1 correspond to the mean of these weekly analysis, with the standard deviation reflecting the variability across weeks.
Comment 8: Why were primiparous and multiparous cows analyzed separately? Could parity be included as a variable in the statistical model?
Response 8: Regarding this question, it was indeed a point of discussion during our data analysis. Initially, we considered analyzing primiparous and multiparous cows together — in fact, we performed this analysis using the following model (general example):
Yᵢⱼₖ = μ + Táµ¢ + Pâ±¼ + εᵢⱼₖ
Where:
Yᵢⱼₖ is the response variable,
Táµ¢ is the fixed effect of treatment,
Pâ±¼ is the fixed effect of parity,
εᵢⱼₖ is the residual error associated with each experimental unit.
However, due to the clear biological differences between parity groups —their response to treatment, their different physiological status during the transition period and the different productive parameters (milk yield, body weight, etc.) — we determined that analyzing them separately would provide more accurate and meaningful interpretations. Although the same treatments were applied and the same outcomes measured in both groups, the transition period presents different metabolic and physiological challenges for primiparous cows compared to multiparous cows. Based on these considerations, the research team agreed to conduct the analysis by parity group.
Comment 9: Is a sample size of only three cows per treatment sufficient to assess treatment effects reliably?
Response 9: We acknowledge the concern regarding the limited sample size. Based on the expected treatment differences and the variability observed in similar studies, we determined that using three cows per treatment was the minimum required to detect effects under our experimental conditions. Although we always aim to include more animals, we faced practical limitations due to the high cost and logistical complexity of working with ruminally cannulated cows. It is worth noting that previous studies conducted by our group have successfully employed similar designs using a total of three cannulated cows per treatment, producing reliable and publishable results (e.g. D.A. Mattiauda, S. Tamminga, M.J. Gibb, P. Soca, O. Bentancur, P. Chilibroste. Restricting access time at pasture and time of grazing allocation for Holstein dairy cows: Ingestive behavior, dry matter intake and milk production (Livestock Science)). To improve the robustness of our data, we replicated sampling on two consecutive days, obtaining multiple samples per cow.
Results
Comment 10: Data on DMI and body condition score (BCS) were not presented in the tables. Could these be included?
Response 10: We appreciate the suggestion. Initially, we included DMI and BCS data in the same table as the performance results. However, given the number of effects tested, adding these variables would have significantly increased the size and complexity of the table, making it more difficult to read and understand. After discussion among the co-authors, we decided that presenting these results in the text would allow for clearer and more concise interpretation, while maintaining the overall readability of the manuscript.
Comment 11: Why were ruminal fermentation evaluations split at 7 days in milk (DIM) but combined for 30 and 60 DIM?
Response 11: The first fermentation evaluation, conducted at −7 days relative to parturition, was analyzed separately due to substantial differences in both the feeding strategy and the physiological status of the cows during the prepartum period. To enhance the biological interpretation of the data, this time point was evaluated independently. Consistent with the analysis of other response variables, the prepartum and postpartum periods were evaluated separately. Postpartum assessments, conducted at 30 and 60 days after calving, were performed under standardized management conditions: cows grazed alfalfa (lucerne) and received a consistent TMR throughout the entire postpartum period.
Comment 12: Was the feed intake of rumen-cannulated cows monitored during fermentation assessments?
Response 12: Thank you for your question. During the prepartum period, we measured the dry matter intake (DMI) of the pen where the cows (including ruminally cannulated cows) were housed. However, we do not have the individual DMI of cannulated cows during this period. During the postpartum period, we estimated the DMI of all cows (including ruminally cannulated cows) for 30 and 60 days of lactation (described in section 2.3 of the manuscript). We hope this has clarified this point.
Reviewer 2 Report
Comments and Suggestions for Authors
This study explores the effects of PHY and MON on the production performance and rumen fermentation of grazing lactating cows. The topic has practical significance, especially in reducing antibiotic use and optimizing pasture management. The experimental design is relatively rigorous, and the data presentation is clear, but some method details and result explanations need further clarification.
PHY and MON only significantly increased milk production in primiparous cows, which the author attributed to their "greater physiological challenges". Is there data to support negative energy balance or higher metabolic stress (such as blood indicators) in primiparous cattle? Relevant analysis or references need to be supplemented.
MON and PHY reduced the concentration of rumen protozoa, but PHY still significantly reduced ammonia nitrogen (NH3-N) at 60 days postpartum, while MON had no such effect. Is it related to the differential inhibition of protozoan populations? Can microbial community changes be validated through molecular biology methods such as 16S rRNA sequencing?
The mechanism by which HY increases branched volatile fatty acids (BCVFA) has not been fully explored. Is it related to the inhibition of specific bacteria or protozoa by phytochemicals? Further analysis is needed in conjunction with existing literature.
What is the internal standard for short-chain fatty acid determination, and it is recommended to describe the method in detail.
Author Response
Comments and Suggestions for Authors – Reply to the Review Report
- Summary
Thank you very much for taking the time to review our manuscript and for your valuable comments and suggestions. Please find our detailed responses below, with the corresponding corrections highlighted in sky blue in the re-submitted file.
Comments and Suggestions for Authors
This study explores the effects of PHY and MON on the production performance and rumen fermentation of grazing lactating cows. The topic has practical significance, especially in reducing antibiotic use and optimizing pasture management. The experimental design is relatively rigorous, and the data presentation is clear, but some method details and result explanations need further clarification.
Comment 1: PHY and MON only significantly increased milk production in primiparous cows, which the author attributed to their "greater physiological challenges". Is there data to support negative energy balance or higher metabolic stress (such as blood indicators) in primiparous cattle? Relevant analysis or references need to be supplemented.
Response 1: Thank you for your question. Regarding this point, an unpublished study by Comesaña et al. (unpublished data) conducted in parallel with the current work, evaluated metabolic adaptation during the transition period. That study found that key metabolites associated with metabolic adaptation and the onset of lactation (e.g., BHB and NEFAs) were elevated in primiparous cows compared to multiparous cows. We have added this contextual information to the discussion section (lines 330-334). However, we are unable to include data from that study directly in this manuscript.
Additionally, previous studies from our group (Chilibroste et al., 2012; Meikle et al., 2018) and others (Eicher et al., 2007) have shown that primiparous cows experience greater physiological challenges during the transition period, particularly under grazing conditions. These findings further support the importance of considering parity in the interpretation of our results.
Comment 2: MON and PHY reduced the concentration of rumen protozoa, but PHY still significantly reduced ammonia nitrogen (NH3-N) at 60 days postpartum, while MON had no such effect. Is it related to the differential inhibition of protozoan populations? Can microbial community changes be validated through molecular biology methods such as 16S rRNA sequencing?
Response 2: Thank you very much for your valuable feedback. While our results indicate a differential effect of PHY and MON on the rumen microbial populations, as evidenced by changes in ruminal fermentation parameters, our study specifically quantified changes in protozoal population counts. We agree that validating these findings using molecular biology techniques would strengthen the conclusions. In fact, rumen samples were collected and preserved during the study for this purpose and are currently being stored for subsequent DNA extraction, sequencing, and microbial community analysis.
Comment 3: The mechanism by which PHY increases branched volatile fatty acids (BCVFA) has not been fully explored. Is it related to the inhibition of specific bacteria or protozoa by phytochemicals? Further analysis is needed in conjunction with existing literature.
Response 3: Regarding the relationship between PHY and branched-chain volatile fatty acids (BCVFA), the underlying mechanisms are still not fully understood. In our study, we observed a diet-related response, where PHY supplementation increased BCVFA concentrations during the prepartum period. As noted by Firkins et al. (2024), BCVFA are derived from the catabolism of branched-chain amino acids (leucine, isoleucine, and valine) present in the diet. Their studies also reported interactions between BCVFA concentrations and ruminal ammonia levels in dairy cows. In our case, the prepartum diet differed substantially from the postpartum diet, particularly in crude protein content and ruminally degradable protein (RDP), which may have influenced BCVFA dynamics. A limitation in interpreting this result is that many studies evaluating the effects of phytochemicals on rumen fermentation do not measure BCVFA. For example, among 19 studies that assessed the impact of PHY on ruminal fermentation in dairy cattle, only 9 reported BCVFA concentrations, and just 3 observed an increase. Additionally, as reported by Firkins et al. (2024), BCVFA concentrations in the rumen are influenced by both their production and utilization. Cellulolytic bacteria are among the primary consumers of BCVFA, as discussed in lines 367- 374. In a parallel study conducted during the prepartum period, we observed a decrease in organic matter degradability, which may suggest reduced cellulolytic bacterial activity or abundance, and consequently, lower BCVFA utilization. This potential decline in BCVFA utilization could explain their accumulation in the rumen under PHY supplementation during the prepartum period. However, we cannot confirm this hypothesis, as 16S rRNA sequencing data were not available for this study.
Comment 4: What is the internal standard for short-chain fatty acid determination, and it is recommended to describe the method in detail.
Response 4: We appreciate the comment regarding the analytical procedure used for the quantification of volatile fatty acids (VFA). In this study, no internal standard was employed, as the samples were neither extracted nor derivatized; they were simply centrifuged and filtered prior to direct injection into the gas chromatography (GC) system. Volatile fatty acid concentrations were quantified using external calibration curves prepared from pure analytical standards (acetic, propionic, butyric acids, among others; Merck, Germany) at known concentrations. These calibration curves were generated under the same analytical conditions as the samples and exhibited high linearity (R² > 0.99). Instrument performance and analytical reproducibility were monitored through regular injections of external standards throughout the analytical sequence. Although the use of internal standards is commonly recommended to account for potential variability in sample management and injection, previous studies have demonstrated that consistent and reproducible quantification of VFAs can be achieved without them, provided that external calibrations, consistent sample processing, and rigorous quality controls are applied. The GC system used in this study was equipped with an automatic injector, and calibration standards were analyzed daily alongside the samples. In response to this comment, we have improved the description of the VFA analytical methodology in the revised manuscript, specifically in lines 205 to 215.
Reviewer 3 Report
Comments and Suggestions for Authors
Congratulations on an ambitious and interesting study. There is much to commend (especially the overall attention to presentation and detail on diets) but there are some matters of concern.
- i) The assertion that the PHY differ from monensin in antimicrobial resistance (AMR) cannot be sustained. Your introduction includes a somewhat bizarre anomaly highlighted by your paper. Simply, as you note in several places the PHY similar to monensin act as antimicrobial agents. It is not, therefore, sustainable to assert that these provide reduced risk of AMR - it is possible that the PHY present greater risk. It is critical that the confused position on risk of PHY and AMR not be repeated in your paper. This will require changes to introduction and discussion.
- ii) It is worth noting in the context that it is only recently that monensin is not available for cattle in the EU and the reason for unavailability relates to supplementation method failure not AMR.
iii) The other most significant concern with the paper is the statistical analysis - I refer you to St Pierre on design and analysis of pen studies. Your unit of interest is the pen and unit of measurement is the cow. Sample sizes for pen studies are usually much larger than you provide. You may need to re-analyze some of the findings. You should describe how your sample size was determined. This comment does not apply to your rumen fistulate analysis.
St-Pierre, 2007 N.R. St-Pierre Design and analysis of pen studies in the animal sciences
- Dairy Sci., 90 (2007), pp. E87-E99
Minor comments
The high quality of presentation obviated many minor comments.
Please reference what sources you used to derive energy estimates from feeds - some are familiar, others not.
It is not necessary or probably useful to comment on the difference between 30d and 60 d measures
Please look at the legend for the rumen analysis figures - are these the 30d and 60d figures
Author Response
Comments and Suggestions for Authors – Reply to the Review Report
- Summary
Thank you very much for reviewing our manuscript and for generating such a valuable discussion. Please find below our detailed responses to your comments, along with the corresponding revisions highlighted in the re-submitted file. All changes made to the manuscript are highlighted in green.
Congratulations on an ambitious and interesting study. There is much to commend (especially the overall attention to presentation and detail on diets) but there are some matters of concern.
Comment 1: The assertion that the PHY differ from monensin in antimicrobial resistance (AMR) cannot be sustained. Your introduction includes a somewhat bizarre anomaly highlighted by your paper. Simply, as you note in several places the PHY similar to monensin act as antimicrobial agents. It is not, therefore, sustainable to assert that these provide reduced risk of AMR - it is possible that the PHY present greater risk. It is critical that the confused position on risk of PHY and AMR not be repeated in your paper. This will require changes to introduction and discussion.
Response 1: Thank you for this valuable comment. It has contributed to improving the rigor of our manuscript. We reconsidered our approach and revised both the introduction and discussion sections accordingly. Specifically, we updated (lines 37–43) in the introduction to emphasize the potential role of phytochemicals in a context where the use of antibiotics is increasingly questioned by the public, and consumers concern over food production practices continues to grow. In the discussion section (lines 420-423), we now highlight that the phytochemical mixture used in this study likely contains compounds with diverse antimicrobial modes of action. This diversity may reduce the likelihood of rapid microbial adaptation compared to additives with a single mechanism. Nonetheless, we acknowledge that microbial adaptation remains a relevant concern for all such feed additives, including those evaluated in this study.
Comment 2: It is worth noting in the context that it is only recently that monensin is not available for cattle in the EU and the reason for unavailability relates to supplementation method failure not AMR.
Response 2: Thank you for your comment. In line with the previous question, we have revised the manuscript to provide a clearer explanation of this point, aiming to improve the reader’s understanding of why monensin is not currently authorized for use in the European Union.
Comment 3: The other most significant concern with the paper is the statistical analysis - I refer you to St Pierre on design and analysis of pen studies. Your unit of interest is the pen and unit of measurement is the cow. Sample sizes for pen studies are usually much larger than you provide. You may need to re-analyze some of the findings. You should describe how your sample size was determined. This comment does not apply to your rumen fistulate analysis.
St.-Pierre, 2007. N.R. St-Pierre Design and analysis of pen studies in the animal sciences J. Dairy Sci., 90 (2007), pp. E87-E99
Response 3:
We greatly appreciate your thoughtful comments regarding the experimental design— as this is a relevant and evolving topic within grazing-based ruminant research. During the postpartum period, a mixed feeding strategy was implemented. In the mornings, all cows assigned to each treatment grazed together on the same paddock. In the afternoons, they were separated by parity and treatment to receive the total mixed ration (TMR) supplementation. Each group consisted of either 12 multiparous or 8 primiparous cows per treatment.
From While we acknowledge that, the pen could be viewed as the experimental unit- since it is the physical entity to which a treatment is applied independently and from which responses are measured- we believe it is important to consider the broader context. As noted by Detmann et al. (2016) in Considerations on Research Methods Applied to Ruminants under Grazing Conditions, experimental design in pasture-based systems must also consider biological, nutritional, and behavioral dynamics to ensure external validity and applicability of experimental outcomes to commercial, real-world production systems.
Striving for a balance between statistical rigor and biological relevance, we opted for a treatment implementation aligned with typical pasture-based dairy systems in Uruguay. This approach was also the only feasible way to apply the treatments under grazing conditions. While treatments were applied at the group level, all cows within a group received the same treatment under uniform management protocols. To ensure consistency and minimize confounding effects we implemented the following measures:
- Cows were separated by parity (primiparous and multiparous) within treatment groups to control for physiological and behavioral differences that could affect responses.
- In the pasture, each group was offered 28 kg of dry matter per cow per day—more than three times the expected intake—to minimize competition and ensure ad libitum access.
- Each cow was allocated 1 meter of feed bunk space to reduce competition and ensure equal access to the TMR.
- All TMR batches were closely monitored during mixing to ensure formulation accuracy and homogeneity.
- Weekly feed evaluations of the feed mixture were conducted to confirm consistent delivery of the intended treatment.
- Animal behavior was monitored via video to ensure compliance with the procedure and detect any behavioral abnormalities.
It is worth noting that all measurements were taken from the individual animal. This approach is consistent with grazing-based experimental designs successfully implemented by the dairy research team at EEMAC, Facultad de Agronomía, Paysandú, Uruguay, where group-level treatment application was applied under pasture-based conditions (e.g., Mattiauda et al., 2013; Méndez et al., 2023; Pons et al., 2023; Mendina et al., 2024). It also aligns with studies published in MDPI journals, such as de Melo et al. (2024), who employed a similar design with grazing beef heifers (Veterinary Sciences).
We hope this explanation clarifies the rationale behind our methodological choices, and demonstrates the measures taken to ensure the integrity and applicability of our findings. We look forward to receiving your further comments and suggestions.
Minor comments
The high quality of presentation obviated many minor comments.
Comment 4: Please reference what sources you used to derive energy estimates from feeds – some are unclear.
Response 4: We incorporated these changes in lines 172 to 180, where we included the references for the equations used to estimate energy values from feeds. Specifically, we used the equations from Rohweder et al. (1984) to calculate the digestible energy (DE) of pasture. In addition, equations from NRC (2001) were used to estimate metabolizable energy (ME) and net energy for lactation (NEL).
Comment 5: It is not necessary or probably useful to comment on the difference between 30d and 60d measures.
Response 5: Thank you for your recommendation. In this case, we believe that the observed differences—particularly the increase in dry matter intake (DMI) from period 30 to 60, as expected—as well as the corresponding increase in organic matter intake and its interaction with the additives, are a relevant point of the article and should be analyzed. Under our experimental conditions, these findings provide important insights and justify further discussion in the manuscript.
Comment 6: Please look at the legend for the rumen analysis figures – are these the 30d and 60d figures?
Response 6: Yes, exactly. These figures illustrate the rumen fermentation patterns, and the main volatile fatty acid (VFA) concentrations observed during ruminal sampling periods (30 and 60 days).
References
Detmann, E.; Paulino, M.F.; Mantovani, H.C.; Valadares Filho, S.C.; Sampaio, C.B.; Souza, M.A.; Lazzarini, I.; Detmann, K.S.C. Considerations on research methods applied to ruminants under grazing conditions. Braz. J. Anim. Sci. 2016, 45, 592–602. https://doi.org/10.1590/S1806-92902016000600010
Mattiauda, D.A.; Tamminga, S.; Gibb, M.J.; Soca, P.; Bentancur, O.; Chilibroste, P. Restricting access time at pasture and time of grazing allocation for Holstein dairy cows: Ingestive behaviour, dry matter intake and milk production. Livest. Sci. 2013, 152, 53–62. https://doi.org/10.1016/j.livsci.2012.12.010.
Medina, G.; Damián, J.P.; Meikle, A.; Adrien, M.L. Metabolic adaptation to lactation of dairy cows in two contrasting facilities involving partial confinement plus grazing or total confinement. Anim. Prod. Sci. 2024, 64, 10. https://doi.org/10.1071/AN23383.
Melo, L.P.; Rennó, L.N.; Detmann, E.; Paulino, M.F.; da Silva Júnior, R.G.; Ortega, R.M.; Sotelo Moreno, D. Effect of Supplementation Plans and Frequency on Performance and Metabolic Responses of Grazing Pregnant Beef Heifers. Vet. Sci. 2024, 11, 506. https://doi.org/10.3390/vetsci11100506.
Méndez, M.N.; Grille, L.; Mendina, G.R.; Robinson, P.H.; Adrien, M.D.L.; Meikle, A.; Chilibroste, P. Performance of Autumn and Spring Calving Holstein Dairy Cows with Different Levels of Environmental Exposure and Feeding Strategies. Animals 2023, 13, 1211. https://doi.org/10.3390/ani13071211
NRC. Nutrient Requirements of Dairy Cattle, 7th ed.; National Academies Press: Washington, DC, USA, 2001; ISBN 978-0309069977
Pons, M.V.; Adrien, M.L.; Mattiauda, D.A.; Breijo, M.A.; Meikle, A.; Chilibroste, P.; Damián, J.P. Welfare of dairy cows in mixed feeding systems under two different conditions of confinement: Behavioral, biochemical and physiological indicators. Appl. Anim. Behav. Sci. 2023, 265, 105995. https://doi.org/10.1016/j.applanim.2023.105995.
Rohweder, D.W. Estimating forage hay quality. In: Proc. Natl. Alfalfa Hay Quality Testing Workshop; Natl. Alfalfa Hay Quality Committee: Chicago, IL, USA, 1984; pp. 31–37.
Round 2
Reviewer 1 Report
Comments and Suggestions for Authors
The authors have addressed some of the questions and incorporated the suggested changes. It is recommended that the authors present the hypothesis first, followed by the study objectives.
Author Response
Comments and Suggestions for Authors – Reply to the Review Report
- Summary
The authors thank the reviewer for their positive comments, which helped improve the manuscript. Below are answers to each reviewer's question, which are highlighted in yellow in the new version of the manuscript
- Comments 1:
The authors have addressed some of the questions and incorporated the suggested changes. It is recommended that the authors present the hypothesis first, followed by the study objectives.
- Response 1
Thank you for your comment. As suggested, we have revised the Introduction to present the hypothesis prior to the study objectives, as shown in lines 76–83.
Lines 76 to 83: “We hypothesized that, compared with the control group, phytochemical supple-mentation improves performance and physiological adaptation to parturition by modulating the ruminal microbial population and fermentation in a manner similar to monensina, thereby improving nutrient utilization. This study aimed to evaluate the effects of a phytochemical mixture and monensin on DMI, body condition score (BCS), MY and milk composition in primiparous and multiparous dairy cows, and ruminal fermentation in primiparous dairy cows, during the transition and early lactation period in a mixed grazing system.”
Reviewer 3 Report
Comments and Suggestions for Authors
I thank the authors for the detailed response on study design - you went to a lot of effort to make this rigorous. None the less, the study is pseudoreplicated - the unit of interest is the pen. You must state this flaw clearly. Pasture studies are always challenging in trying to ensure effective replication but there were ways you could have avoided this. I would normally reject pseudoreplicated papers but see the strength in your approach. Perhaps an expansion of detail on methods would allow the strength to be obvious while clearly stating the flaw.
I have a problem with your positioning of the product. If the product works it must have antimicrobial actions - so why assert from an untested perspective that it will be more or less likely than monensin to produce AMR.
You must be clear that the product is an antimicrobial agent as evidenced by your observed changes in metabolism. This needs to be stated in the introduction and discussion. It is important to provide a similar clarity and scrutiny to all antimicrobial agents.
Author Response
Comments and Suggestions for Authors – Reply to the Review Report
- Summary
Thank you very much for reviewing our manuscript and for generating such a valuable discussion. Please find below our detailed responses to your comments, along with the corresponding revisions highlighted in yellow.
- Comments 1:
I thank the authors for the detailed response on study design - you went to a lot of effort to make this rigorous. None the less, the study is pseudo replicated - the unit of interest is the pen. You must state this flaw clearly. Pasture studies are always challenging in trying to ensure effective replication but there were ways you could have avoided this. I would normally reject pseudo replicated papers but see the strength in your approach. Perhaps an expansion of detail on methods would allow the strength to be obvious while clearly stating the flaw.
- Response 1:
Thank you for you valuable comment. We have revised the Materials and Methods section to provide a more detailed explanation of the experimental design (lines 120 to 130 and 450 to 457) including the rationale behind our decisions and the inherent limitations of the study. We have also clarified the procedures we followed to justify the use of the individual cow as the experimental unit, despite the potential concerns of pseudo-replication.
Lines 120 to 130: “Postpartum, cows were grouped by category and treatment, with each group consisting of either eight primiparous or twelve multiparous cows from the same treatment. This grouping strategy was determined by the experimental conditions and designed to reflect commercial dairy systems in Uruguay. All variables of interest were measured at the individual level; therefore, in this study, the individual cow was considered as the experimental unit. To minimize competitive interactions during the grazing period, cows were offered 28 kg of DM/cow/day of pasture. While housed in the compost barn, cows were separated by parity to prevent confounding effects during treatment administration. To ensure consistent delivery of the TMR, a daily monitoring protocol was implemented, and each cow was allocated 1 meter of feed bunk space to ensure unrestricted access to the assigned diet.”
Lines 450 to 457: “While the use of the individual animal as the experimental unit may be considered a limitation in dairy studies where animals share a common environment (e.g., a pen), this approach may be valid when the effects of the group or shared environment are adequately controlled (as previously described). Furthermore, it offers greater sensitivity for detecting biologically significant individual variations relevant to the study objectives. However, these results should be interpreted with caution, as the animal was used as the experimental unit despite treatments being applied at the group level. Therefore, further studies are warranted to confirm our findings.”
- Comments 2:
I have a problem with your positioning of the product. If the product works it must have antimicrobial actions - so why assert from an untested perspective that it will be more or less likely than monensin to produce AMR
- Response 2:
Thank you for this important observation. We have revised the Simple Summary (lines 1 to 6) and the Introduction (lines 46 to 49) to adjust the positioning of the product accordingly. In the Discussion section, lines 429 to 435 we now clearly state that concerns regarding antimicrobial resistance (AMR) associated with monensin could similarly apply to phytochemicals, given their antimicrobial properties. We acknowledge that both products may exert selective pressure on microbial populations, and this potential risk must be considered equally for both.
Lines 1 to 6: “The use of antibiotics in livestock farming has raised significant public concern and is increasingly being questioned, particularly in the case of monensin, one of the most widely used additives. As a result, there is growing interest in finding alternatives to improve animal health and production efficiency. Phytochemicals are promising natural compounds with greater public acceptance; however, limited information is available regarding their application in dairy grazing systems during the transition period.”
Lines 46 to 49: “These compounds exhibit antimicrobial similar to those of monensin, as well as antioxidant properties, and are promising feed additives capable of enhancing rumen fer-mentation, improving feed efficiency, and promoting animal health [7].”
Lines 429 to 435: “Although the PHY mixture also exerts antimicrobial activity, it effects may involve multiple mechanisms, including disruption of microbial membranes, enzyme inhibition, and interference with cell signaling [7, 11]. These diverse actions could potentially limit microbial adaptation during the evaluation period. However, with prolonged exposure, microbial adaptation to these mechanisms may still occur, raising concerns similar to those associated with monensin and highlighting the need for further investigation.”
- Comments 3:
You must be clear that the product is an antimicrobial agent as evidenced by your observed changes in metabolism. This needs to be stated in the introduction and discussion. It is important to provide a similar clarity and scrutiny to all antimicrobial agents.
- Response 3:
Thank you for your comment. We have clarified that the phytochemical mixture exhibits antimicrobial activity, similar to monensin and other antimicrobial agents, and that this activity is linked to the observed changes in ruminal metabolism. We have included a statement highlighting this point in both the Introduction and Discussion sections to ensure greater clarity and consistency with the characterization of antimicrobial agents. We hope these changes sufficiently clarify that phytochemicals possess antimicrobial activity, acting in a manner similar to other antimicrobial agents.
Lines 46 to 48: “These compounds exhibit antimicrobial similar to those of monensin, as well as antioxidant properties, and are promising feed additives capable of enhancing rumen fermentation, improving feed efficiency, and promoting animal health [7].”
Lines 429 to 432: “Although the PHY mixture also exerts antimicrobial activity, it effects may involve multiple mechanisms, including disruption of microbial membranes, enzyme inhibition, and interference with cell signaling [7, 11].”